# The WC and CrC Coatings Deposited from Carbonyls Using PE CVD Method—Structure and Properties

**DOI:** 10.3390/ma16145044

**Published:** 2023-07-17

**Authors:** Marianna Trebuňová, Daniel Kottfer, Karol Kyziol, Mária Kaňuchová, Dávid Medveď, Róbert Džunda, Marta Kianicová, Lukáš Rusinko, Alena Breznická, Mária Csatáryová

**Affiliations:** 1Department of Biomedical Engineering and Measurement, Faculty of Mechanical Engineering, Technical University of Košice, Letná 9, 042 00 Košice, Slovakia; marianna.trebunova@tuke.sk; 2Department of Mechanical Technologies and Materials, Faculty of Special Technology, Alexander Dubček University of Trenčín, Ku Kyselke 469, 911 06 Trenčín, Slovakia; marta.kianicova@tnuni.sk (M.K.); lukas.rusinko@tnuni.sk (L.R.); alena.breznicka@tnuni.sk (A.B.); 3Department of Physical Chemistry and Modelling, Faculty of Materials Science and Ceramics, AGH University of Science and Technology, A. Mickiewicza 30 Av., 30 059 Kraków, Poland; kyziol@agh.edu.pl; 4Raw Materials Processing Department, Institute of Mountainous Sciences and Environmental Protection, Faculty of Mining, Ecology, Process Control and Geotechnology, Technical University of Košice, Park Komenskeho 19, 043 84 Košice, Slovakia; maria.kanuchova@tuke.sk; 5Institute of Materials Research, Slovak Academy of Sciences, Watsonova 47, 040 01 Košice, Slovakia; dmedved@saske.sk (D.M.); rdzunda@saske.sk (R.D.); 6Department of Physics, Mathematics and Technologies, Faculty of Humanities and Natural Sciences, University of Presov, Ul. 17 Novembra 1, 080 01 Prešov, Slovakia; maja@unipo.sk

**Keywords:** WC coating, CrC coating, W hexacarbonyl, Cr hexacarbonyl, plasma method, surface properties

## Abstract

This article presents a comparative study of WC and CrC coatings deposited by the plasma-enhanced chemical vapor method using the hexacarbonyls of W and Cr as precursors. The measured thicknesses of the WC and CrC coatings are equal to ca. 1.5 µm. The WC coating consists of microcolumns with a conical end, with gaps between the microcolumns up to approximately 100 nm, and their structure is formed by nanoparticles in the shape of globules with a diameter of up to 10 nm. In the case of the CrC coating, a cauliflower structure with gaps ranging from 20 to 100 nm was achieved. The diameter of cauliflower grains is from 50 nm to 300 nm. The C content in the WC and CrC coating is 66.5 at.% and 75.5 at.%. The W content is 1.4 at.% and the Cr content in the CrC coating is 1.2 at.%. The hardness and Young’s modulus of the WC coating are equal to 9.2 ± 1.2 GPa 440.2 ± 14.2 GPa, respectively. The coefficients of friction and wear volume of the WC coating are equal to 0.7 and −1.6 × 10^6^/+3.3 × 10^6^ µm^3^, respectively. The hardness and Young’s modulus of the CrC coating are 7.5 ± 1.2 GPa and 280 ± 18.5 GPa, respectively. The coefficients of friction and wear volume of the CrC coating are 0.72 and −18.84 × 10^6^/+0.35 × 10^6^ µm^3^, respectively.

## 1. Introduction

Wolfram carbide (WC) and chromium carbide (CrC) are exceptionally hard layered materials (H, WC ca. 42 GPa [1,2,3], CrC ca. 25 GPa [4,5]) with good abrasion resistance that are characterized by a low coefficient of friction (COF for WC of 0.22 [1,3] and CrC of 0.1–0.2 [6]). Therefore, these materials can be deposited on functional surfaces of mechanical components as thin layers in order to increase their long life. The melting point of WC is 2870 °C and that of CrC is 1414 °C [7], and this is the reason why WC and CrC coatings are used in the industry.

WC and CrC coatings are very often deposited using the physical vapor deposition (PVD) method, which includes magnetron sputtering (MS) [1,2,3,4,5,6,8,9,10,11,12,13,14,15,16], evaporation [6], high target utilization sputtering (HiTUS) [17], and the application of high-power impulse magnetron sputtering (HiPIMS) [5,9]. Usually, by using unconventional methods, the temperature of the deposition process does not exceed 400 °C. Sometimes, chemical vapor deposition techniques [18,19,20] are also used. In this case, the disadvantages include high deposition temperatures, which can reach as much as 1100 °C. However, the deposition temperature can be significantly lowered (to under 400 °C) using ionized working gas and applied low-pressure plasma conditions. These techniques are also named the plasma-enhanced (PE), plasma-activated (PA), or plasma-inducted (PI) chemical vapor deposition (PECVD, PACVD, or PICVD) methods. Generally, in order to obtain the WC and CrC coatings using chemical and physical vapour deposition methods, wolfram carbonyl [19,21,22,23,24,25], chromium carbonyl [23,26,27,28,29,30,31,32,33,34,35,36,37,38], and molybdenum carbonyl [19,29], which sublime in the reactor at low temperatures, can be used as precursors. There is currently little research on WC and CrC PECVD coatings deposited from carbonyls of W and Cr.

In recent years, many researchers have studied the technology design and properties of these coatings, and have conducted this research on multilayered systems, too. For example, Huang et al. [1] have studied the relationship of the mechanical and tribological properties of CrC/a-C:H coatings deposited with anode-assisted reactive magnetron sputtering combined with DC-pulsed plasma-enhanced chemical vapor deposition. They have controlled the flow rate of the additive gas C_2_H_2_ from 0 sccm up to 30 sccm. By such alteration, the content of C in the coating from 12 to 58 at.% has been reached. The maximum value of hardness ca. 16.5 GPa has been measured on a CrC coating, which contained 38 at.% of C. The minimum value of hardness (6.9 GPa) has been obtained in the same coating, which only varied in the C content (12 at.%).

Tillmann et al. [5] researched the impact of the flow rate of additive gas C_2_H_2_ on the at. content of C, and the mechanical and tribological properties of the CrC coating were sequentially deposited using HiPIMS, mid-frequency magnetron sputtering (mfMS), hardness, and coefficient of friction. Their obtained measurements of C are equal to 50, 65, and 77 at.%. As for hardness, the measurements are 13.8 ± 2.4, 13.7 ± 2.0 and 12.3 ± 3.1 GPa, and the coefficient of friction measurements are 0.45 ± 0.02, 0.16 ± 0.01 and 0.16 ± 0.02.

Anderson et al. [16] evaluated the effect of at. C in CrC DC magnetron sputtered coating on hardness, and the at. content of C in the evaluated coating reached was from 85% to 25%, with a measured hardness from 6.9 ± 0.8 GPa to 10.6 ± 1.1 GPa. However, they stated that hardness increased in direct proportion with the relative amount of C-Cr bonds.

The pressure of W(CO)_6_ and Cr(CO)_6_ steams is important for the technologies of these coatings. It is also dependent on the temperature, and is defined by the following equations [30,31,32]:log *p* = 10.65 − 3872/*T*, in the case of W(CO)_6_(1)
log *p* = 10.63 − 3285/*T*, in the case of Cr(CO)_6_(2)
where *p* is pressure of the steams [kPa], and where *T* is absolute temperature [K].

It can be concluded that the pressure of W(CO)_6_ steams in temperatures ranging from 20 °C to 40 °C comes from an interval ranging from 2.7 Pa up to 19 Pa. As for the pressure of Cr(CO)_6_ steams, they range from 17 Pa up to 105 Pa. This allows a relatively high velocity of sublimation of carbonyls to be obtained on the condition that the working pressures in the vacuum chamber of the adjusted PVD apparatus range from 0.01 Pa to 10 Pa. In these conditions, the Cr(CO)_6_ decomposes freely at a temperature of 100 °C (so, importantly, the decomposition temperature of W(CO)_6_ is 170 °C) [31,32]. The decomposition can be accelerated at a temperature of 135 °C [30].

Therefore, the aim of this article is to research the mechanical, tribological, and structural properties of WC/C and CrC/C coatings deposited using the PECVD method using the hexacarbonyls of W and Cr as precursors with the same value of other technological parameters. The obtained results will be compared with each other and with the results of other published works. Research of the mentioned properties and their mutual comparison can contribute to further research on the deposition of WC and CrC coatings using hexacarbonyls of W and Cr as precursors.

## 2. Materials and Methods

### 2.1. Sample Preparation

The two types of substrate are used in the experiment. The first substrate, which is to evaluate the thickness and structure of obtained coatings, consists of monocrystal Si with dimensions of 20 mm × 15 mm × 1 mm. The second substrate, which is to evaluate selected mechanical and tribological properties, is made of C45 (STN 412050) steel in the shape of a disc with a diameter of 52 mm and a thickness of 3 mm. The chemical composition (wt. %) of steel substrate consists of 0.42–0.50 C, 0.40 Si, 0.50–0.80 Mn, 0.40 Cr, 0.10 Mo, 0.40% Ni, and 0.035 P, according to the STN 412050 [24]. The substrates are hardened in oil during the heating process that reaches the temperature of 860 °C. Next, the substrates are gradually polished using various diamond pastes with granularity equal to 15 µm, 9 µm, and 3 µm, respectively. Lastly, the samples are polished using diamond paste with a granularity of 1 µm in order to achieve a surface roughness (R_a_) of ca. 12 nm. The samples are then cleaned in acetone using ultrasound for 10 min and are later dried using a stream of warm air for 5 min. Finally, the substrates are placed into a vacuum chamber (on a cathode) and etched in Ar plasma at the pressure of 10^−3^ Pa, where the bias of the holder (U_b_) is −5 kV, the current density is 1 mA·cm^−2^, and time of process is 15 min. The Ar flow in the vacuum chamber is 65 cm^3^·min^−1^ [25,33].

### 2.2. Coating Deposition

The coatings are deposited using a ZIP-12 (NTC New Technology Centre, Košice, SK, Slovakia) apparatus with a sublimation chamber for the placement of the crucible containing carbonyl (Figure 1) by the plasma-enhanced chemical vapor deposition (PECVD) method, which is conducted by applying direct-current electric voltage [25,33]. The vacuum chamber is separated from the sublimation chamber using a butterfly valve so as not to let the carbonyl sublime during the pumping stage.

Argon (working gas) is dosed using a regulation unit Balzers RVG 040, with feedback from Pirani vacuum gauge to a value of pressure in the vacuum chamber. The samples are cleaned using Ar^+^ ions (please see part 2.1 of the manuscript). WC and CrC coatings are deposited using the same value of technological parameters given in Table 1.

The temperature of substrates does not exceed 300 °C and is controlled using Kapton tape (made out of polyimide with a silicone sticking surface on one side). W(CO)_6_ (W hexacarbonyl) and Cr(CO)_6_ (Cr hexacarbonyl) are used as precursors during the deposition process of the WC and CrC coatings, respectively.

Generally, W(CO)_6_ decomposed during deposition to W and 6 CO and then according to the Bourdouard reaction, and it also decomposed to carbon dioxide (2CO → C + CO_2_) [34]. In the case of Cr(CO)_6_ as a precursor, the two mentioned reactions take place in a similar manner, and they finally lead to the creation of a CrC coating on the substrate.

### 2.3. Nanohardness and Young’s Modulus

Nanohardness and Young’s modulus are measured using the instrumental indentation method (Nanoindenter NHT, CSM Instruments, Basel, Switzerland). The measurements are provided using sinus mode with an amplitude equal to 1 mN, an applied load of 20–60 mN, and a frequency of 15 Hz. The values of indentation hardness and indentation modulus are calculated as an average of the maximal values of the indentation curves [25,33].

### 2.4. Coefficient of Friction and Wear

The coefficient of friction of the compared WC and CrC coatings is evaluated using a ball-on-disc method on a HTT tribometer (CSM Instruments, Needham, MA, USA). The parameters of the test are as follows: as a counterpart, a 100Cr6 steel ball with a diameter of 6 mm was used; the path radius was 4 and 6 mm; the velocity of the counterpart was 5 cm/s; the load force was 0.5 N; the ambient temperature was 25 °C; and the sliding distance was 50 m. The coefficient of friction of the coated substrate surface is continually noted down for each test as a function of sliding distance.

The wear of the coated surface is evaluated as a diminishing volume *V*. Firstly, the surface of the profile of the course *S* at four positions along the circumference always swiveled by 90° using a confocal microscope Neox Plu (Sensofar, Barcelona, Spain). The diminished volume *V* (µm^3^) is calculated according to the formula:*V* = 2 × π × *r* × *S*
(3)
where *r* is radius [mm], and where *S* is the surface of the cross-section of the mark after the test [µm^2^].

### 2.5. SEM and XPS Analyses

The surface morphology of the evaluated coatings is observed using an electron microscope, JEOL JSM 7000F (Tokyo, Japan).

In addition, the thicknesses of the coatings are evaluated based on the cross-section of the WC and CrC coatings deposited on the Si substrates.

The chemical composition and selected atomic groups of the tested structures are observed using X-ray photoelectron spectroscopy (XPS). The XPS analysis is performed on an instrument called SPECS (SPECS GmbH, Berlin, Germany), which is equipped with a PHOIBOS 100 SCD and a non-monochromatic X-ray source. The survey surface spectra of samples are measured at 70 eV transition energy and the core spectra are measured at 30 eV at room temperature. The spectra are all obtained at a basic pressure of 1 × 10^−8^ mbar with MgKα excitation at 10 kV (200 W). The data analysis is performed using CasaXPS Version 2.3.17 software (Casa Software Ltd., Teignmouth, UK). A Shirley and Tougaard type of baseline correction is used for all peak fits.

## 3. Results and Discussion

### 3.1. Thickness, Morphology and Chemical Composition

The WC coating consisted of grain-like structures in a columnar shape, as shown in [22,23], where the shape of the end of each such structure appeared as a cone (Figure 2a, see arrows). These grains had a diameter ranging from 0.5 µm up to 2.5 µm, and they diverted from an imaginary line perpendicular to the coated surface by ca. 20°. This can be attributed to the placement of the sample in the vacuum chamber. The measured thickness of the WC coating was ca. 1.5 µm (Figure 2b). Each individual grain was made out of nanograins in the shape of a spherule with a diameter ca. 5 nm (Figure 2c), which is similar to the WC structure published in [22,23]. The structure of the obtained coating was columnar with gaps between grains that are ca. 20 nm wide (Figure 2c), which contrasts with [22,23], where the gaps are not present. This structure was simultaneously similar to a porous, lacunal structure consisting of grains in the shape of columns, which was defined in the Thornton diagram for a fraction of T/T_m_ 0.105 (300 °C/2870 °C) and pressure of 30 mTorr [35]. However, the structure of the evaluated coating was denser, with a significantly larger diameter of the columns. One can also state that the structure of the obtained coating was in good agreement with the Ta, Cr, Nb, and Al coatings deposited using glancing angle deposition (GLAD) while the substrate was rotated continually at the ratio of the temperature of Θ (reduced temperature) = T_s_ (substrate temperature)/T_m_ (melting temperature of coating material) = 0.4 [36,37].

On the other hand, the CrC coating had a cauliflower structure with considerably larger gaps between the individual grains (Figure 3a). The grains of a cauliflower-like shape had a diameter ranging from 20 nm up to 250 nm (Figure 3c), which is in accordance with [20] and akin to the WC structure published in [22,23], where the gaps are significantly narrower. The obtained thickness of the CrC coating was 1.3 µm (Figure 3b). It was also in good agreement with the Ta, Cr, Nb, and Al coatings deposited using GLAD while the substrate was rotated continually at the ratio of the temperature of Θ equal 0.29 (Ts/T_m_, 300 °C/1414 °C) [36,37]. Furthermore, the thickness of the obtained CrC coating (ca. 1.3 µm, Figure 3b) and the thickness of the structure were similar to the structure of a CrC coating deposited using the HiPIMS method [5]. The obtained coating contained ca. 52 at.% C, which was ca. 24.5 at.% Cr less than the coating obtained by us. Moreover, the hardness of the CrC coating obtained by Tillmann et al. [5] was 6 GPa higher when compared to the evaluated CrC coating.

The growth rate of the WC coating was greater when compared to the CrC coating. Obtained coatings were deposited at a temperature of 22 °C, pressure ranging from 3 to 4 Pa. Cr(CO)_6_ decomposed freely at a temperature of 100 °C, and W(CO)_6_ decomposed at temperatures ranging from 20 °C up to 40 °C, which, under the pressure from 3 to 4 Pa, could have caused a higher growth rate and higher thickness of the WC coating in comparison to the CrC coating.

The quantitative analysis of the survey spectra of the sample with WC coating performed using CASAXPS software indicated, at the surface, 32.1 at.% of oxygen, 66.5 at.% of carbon, and 1.4 at.% of tungsten. The fitting of C 1s, O 1s, and W 4f core level spectra was carried out using CASAXPS software. The binding energy of the survey in the whole range of the specimen was illustrated in Figure 4a, which revealed that the major element on the surface layer of the tested sample was carbon.

Figure 4b showed the high-resolution XPS spectra on the binding state of C 1s. The C 1s spectrum of a sample with a carbon layer included three peaks at binding energies 284.04 eV, 284.14 eV, and 287.06 eV, attributed to sp^3^ hybridization (diamond phase), sp^2^ hybridization (graphite phase), and the C=O bond from oxygen of carbon from the carbonyl of W [38].

The O 1s spectra (Figure 4c) showed two groups of oxygen. The binding energy at 531.67 eV and at 533.54 eV belongs to two types of oxygen: the first was oxygen bonded with tungsten and the second was oxygen bonded with carbon [39].

The W 4f spectra (Figure 4d) showed three groups of tungsten. The binding energy at 36.59 eV belongs to a type of tungsten related to oxygen (WO_3_). The binding energy at 32.36 eV and at 30.53 eV belongs to types of tungsten related to carbide WC [40]. The peaks at 37.15, 34.3, 35.29, and 31.23 eV can be ascribed to clear tungsten [41].

Chemical composition of WC coating was evaluated using EDS analysis (Figure 5). EDS spectrum (Figure 5b) from the surface of the coating (Figure 5a) showed higher at% W and lower at% of oxygen. This could influence hardness and COF.

In the case of the CrC coating, the XPS spectra (Figure 6a) clearly showed three peaks from O 1s (531.7 eV), C 1s (284.5 eV), and Cr 2p (576.5 eV) [42]. The quantitative analysis of the survey spectra of the sample with obtained coating indicated, on the surface, 23.3 at.% of oxygen, 76.5 at.% of carbon, and 1.2 at.% of chromium.

As shown in Figure 6b, the C 1s spectrum can be deconvoluted into three peaks at 287.33 eV, attributed to the C=O bond (carbon and oxygen from air); 284.52 eV, which is typically for sp^3^ hybridization (diamond phase); and 283.03 eV, which is assigned to the carbon atoms in sp^2^ hybridization (graphite phase) [43].

As shown in Figure 6c, the O 1s spectrum can be deconvoluted into two peaks at 531.96 eV, attributed to the C=O bond, and at 531.31 eV, attributed to Cr_2_O_3_ [39].

The Cr 2p band could be easily decomposed to constituent peaks at 576.63 eV and 586.57 eV, assigned to oxide chromium with Cr(3+) oxide phase. The peak with the values 574.32 eV and 584.38 eV belongs to the compound with the CrC composition (chromium carbide) [43,44,45,46].

The chemical composition of the CrC coating was evaluated using EDS analysis (Figure 7). EDS spectrum (Figure 7b) from the surface of the coating (Figure 7a) showed higher at% W and lower at% of oxygen. This was similar to the case of the WC and could influence the hardness and the COF.

### 3.2. Hardness and Young’s Modulus

The hardness of the obtained WC coating achieved a value of 9.2 ± 1.2 GPa (Figure 8), and a similar result was observed by Meng et al. [47] (including ca. 78 at.% C). The obtained values of El Mrabet et al. [3] and Czyzniewski [1] were 20.0 ± 2.0 GPa at 69.0 at.% C and 65 at.% C, which is double in comparison to the evaluated WC coating. Park et al. [48] deposited WC coating containing ca. 65 at.% C and measured hardness at ca. 15 GPa, which is 50% greater when compared to the evaluated WC coating.

The obtained WC coating contained 66.5 at.% C and a high content of carbon involved the creation of diamond-like phase (sp^3^ bond, Figure 4b), and it should ensure a high value of surface hardness. The lower value of hardness (9.2 ± 1.2 GPa) can be caused by the structure of the coating, which was made out of gapful nanograins with space between the grains (Figure 2c). On the other hand, the structure of the WC coating did not contain empty spaces, but was formed in cone-shaped endings of the columnar grains, which were slanted towards the coated surface. This can be the cause of the lower value of hardness when compared to the CrC coating (Figure 2 and Figure 3) and with [22,23], which was proved by lower measured values of hardness and Young’s modulus. Young’s modulus of WC and CrC coatings was measured as equal to 440.2 ± 14.2 GPa and 280 ± 18.5 GPa, respectively (Figure 9).

The hardness of the CrC coating (75.5 at.% C) reached a value of 7.5 ± 1.2 GPa (Figure 8), which was ca. 15% less than the obtained WC coating. Concurrently, it was in agreement with results obtained for the CrC coatings using the MS PVD method by Anderson et al. [16], where H_IT_ ca. 7.9 ± 0.9 GPa and E_IT_ ca. 152 ± 10 GPa were observed (with contents of C at 67 at.%).

WC and CrC coatings contained 66.5 and 75.5 at.% C, respectively. Both of these coatings were contained a high amount of C in its hard sp^3^ bond form. The structure of the WC coating was formed by columns with cone-like endings. Nanogaps were situated between the columnar grains (Figure 2c). On the other hand, the CrC coating had a cauliflower structure and contained larger gaps in between its grains, as in the case of the WC coating (Figure 3c). These gaps also created empty spaces with dimensions up to 100 × 200 nm. The surface of the CrC coating was also rougher. This caused a significantly lower value of hardness when compared to [22,23]. This was the reason why the high content of C in its hard sp^3^ bond form did not display a high value of hardness. The structure of the WC coating was denser in comparison to the CrC coating, and this could be the cause of the relatively low measured values of hardness for both the CrC and the WC coatings.

CrC coating contains 23.3 at.% of O. The hardness was measured at 7.5 ± 1.2 GPa, which is in good agreement with Huang et al. [4], who obtained the value of 7.9 GPa for 10 at.% O. With a decrease in at.% of O, the hardness increased [4].

Evaluated WC coating contains ca. 32 at.% of O. The obtained value of hardness was 9.2 ± 1.2 GPa. Abad et al. measured the hardness of ca. 40 GPa with a content of at.% of O ca. 5. With a decrease in content of O (2 at.% of O), the hardness did drastically decrease (ca. 16 GPa) [4].

When the deposition was carried out, the factor that was controlled was whether this temperature had or had not exceeded the value of 300 °C using Kapton tape, which was stuck along the edge of the sample. It can be stated that the deposition temperature could have reached a markedly lower value than 300 °C. Therefore, it can be stated that the cause of the mentioned and described WC and CrC structures was the low kinetic energy of the falling particles of the precursor.

### 3.3. Coefficient of Friction and Wear

The measured values of the COF of the WC coating with content of C at 76.2 at.% were 0.72 and 0.86 (Figure 10). These results corresponded with the measured value of the COF (ca. 0.72, by Czyzniewski [1]), but the content of C in these coatings was 40 at.% C. At 65 at.% C, they have obtained a value of COF = 0.12.

El Mrabet et al. [3] measured the COF equal to 0.72 and 0.80 with a content of C ca. 40 and 38 at.% C, ly.

Voevodin et al. [49] measured the COF of ca. 0.4 in the WC coating with an equal amount of at.% C as we did in this article. On the other hand, the obtained value of the COF by Voevodin et al. [49] was ca. 0.7, and it was measured on a WC coating that contained 35 at.% C.

Park et al. [48] measured COF = 0.65 and 0.13 with 85 and 35 at.% C in the WC coating.

The COF of the evaluated CrC coating was ca. 0.72 and 0.9, which was 10% higher than results obtained by Anderson et al. [16], whose CrC coating contained 85 at.% C. Tillmann et al. [5] measured the COF of ca. 0.46 and 0.16 with 50 a 85 at.% C present in the CrC coating. This can be attributed to having carbon in its graphite form, which acted as a lubricant in the coating. This was also confirmed by the value of the measured hardness ca. 13 GPa.

The obtained high COF values of both WC and CrC coatings can be caused by coatings‘ surface breaking down into hard particles as well as their abrasive effect during the test.

The wear was evaluated as a dwindling volume of the surfaces‘ material after the test (using two radiuses: 4 mm and 6 mm). Profiles of the evaluated coatings‘ marks (Figure 10) were therefore different, and they point towards differing characteristics of wear. The 0-0 plain identified the surface of the coated sample before the test. The area between the curve above the 0-0 plain and the 0-0 plain did set bounds to both the likely repositioned material of the counterpart on the surface of the WC coating (Figure 11a,b) and the repositioned volume wear of the CrC coating (Figure 11c,d).

Scratches did gradually form on the surface of the harder WC coating (Figure 11a,b), and it was entirely possible that parts of the counterparts‘ material did also move (increased value) above the 0-0 axis. Wear of the counter-sample was not evaluated. In the case of the CrC coating, gradual diminishing of the surfaces‘ material took place (cross-section of the footprint below the 0-0 axis, with decreased value). Due to the thickness of the CrC coating being 1.3 µm and the depth of the wear tracks after the test being ca. 2 µm and 5 µm (Figure 11c,d), it can be stated that the WC coating was overtoiled. Simultaneously, a part of the coating was repositioned on the edge of the footprint above the 0-0 plain (increased value).

The obtained values of the decreased and increased volume after the pin-on-disc test were in agreement with the measured values of hardness, and they are included in Table 2.

## 4. Conclusions

The use of hexacarbonyl W and hexacarbonyl Cr as precursors in the process of depositing WC and CrC coatings deposited at the same technological parameters have made it possible to form the following conclusions:i.Given equivalent technological parameters, two different structures of the coating were obtained: a cauliflower structure (CrC coating) and a column structure with cone-like endings of these columns (WC coating).ii.The obtained WC coating deposited using the PECVD method contained 66.5 at.% C, and it was characterized by interesting mechanical and tribological properties: H_IT,_ 9.2 ± 1.2 GPa; E_IT_, 440 ± 19 GPa; and COF, ca. 0.80.iii.The obtained CrC coating deposited using the PECVD method contained 76.2 at.% C, and it was characterized by interesting mechanical and tribological properties: H_IT_ = 7.5± 1.2 GPa; E_IT_ = 280 ± 23 GPa; and COF = 0.72 and 0.90.iv.Given the equal technological parameters of the deposition, the obtained thickness of the WC and the CrC coatings were 1.5 µm a 1.3 µm. The growth rate of the WC coating was higher when compared to CrC due to the decomposition temperature of W(CO)_6_ at pressure, ranging from 3 to 4 Pa that ranges from 20 °C to 40 °C. In the case of Cr(CO)_6_, it was 100 °C. The evaluated coatings were deposited at a temperature of ca. 22 °C and with pressure ranging from 3 to 4 Pa.v.The WC coating contained 66.5 at.% C, whereas the CrC coating contained 75.5 at.% C. Both coatings showed a high content of C in its hard sp3 bond form. The structure of the WC coating was made out of columns with cone-like endings. Nanogaps were situated between the columnar grains. On the other hand, the CrC coating had a cauliflower structure, and it had larger gaps between individual grains when compared to the WC coating. The mentioned gaps could also be a cause of the relatively low measured values of hardness.vi.The evaluated coatings contained a sufficient amount of at.% groups WO, WC, Cr_2_O_3_, and CrC. Their sufficient amounts did not affect hardness and Young´s modulus due to the aforementioned structures. This could be attributed to the low temperature of the deposition.vii.The COF of the WC and CrC coatings was 0.72 and 0.86 and 0.72 and 0.9, respectively. In the case of the CrC coating, hard particles, such as CrC, Cr_2_O_3_, and sp^3^C states, were observed. These particles can create smaller particles between the coating and the counter-sample, which can lead to wear. In the case of the WC coating, scratches did form on its surface, and the transition of a part of the counter-sample (steel ball 100Cr6) material onto the wear tracks‘ surface took place.viii.The different temperature of sublimation of Cr and W carbonyls affected the velocity of sublimation and the growth rate of the evaluated coatings. The growth rate of the WC coating was 8.5 nm/min, whereas for the CrC coating, it was 7.5 nm/min.ix.Based on the measured properties, a better WC coating is shown.

Further research will be focused at the impact of bias on the mechanical, tribological, and structural properties of CrC and WC coatings deposited using the PECVD method.

## Figures and Tables

**Figure 1 materials-16-05044-f001:**
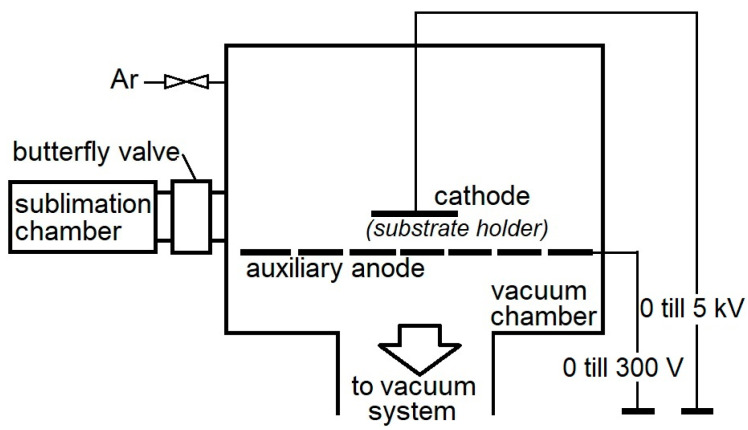
Scheme of PECVD system (ZIP 12 type) with the sublimation chamber used during the deposition of the WC and CrC coatings.

**Figure 2 materials-16-05044-f002:**
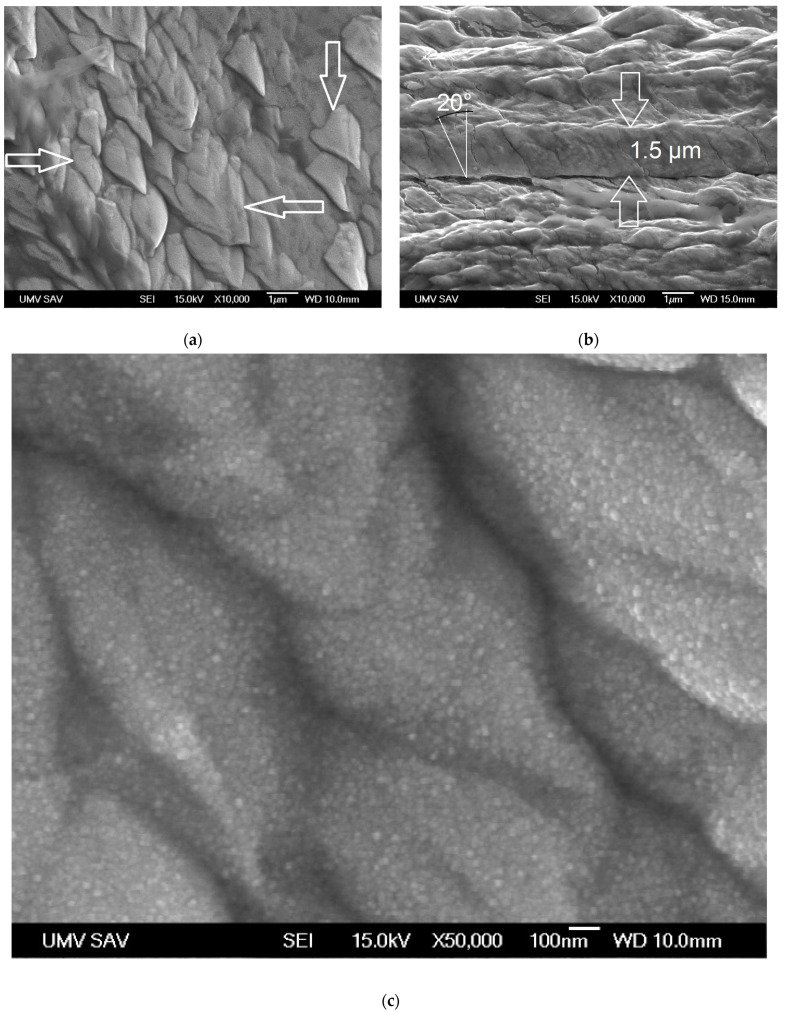
SEM images of the WC coating: (**a**) surface morphology; (**b**) cross-sectional view; (**c**) microstructure.

**Figure 3 materials-16-05044-f003:**
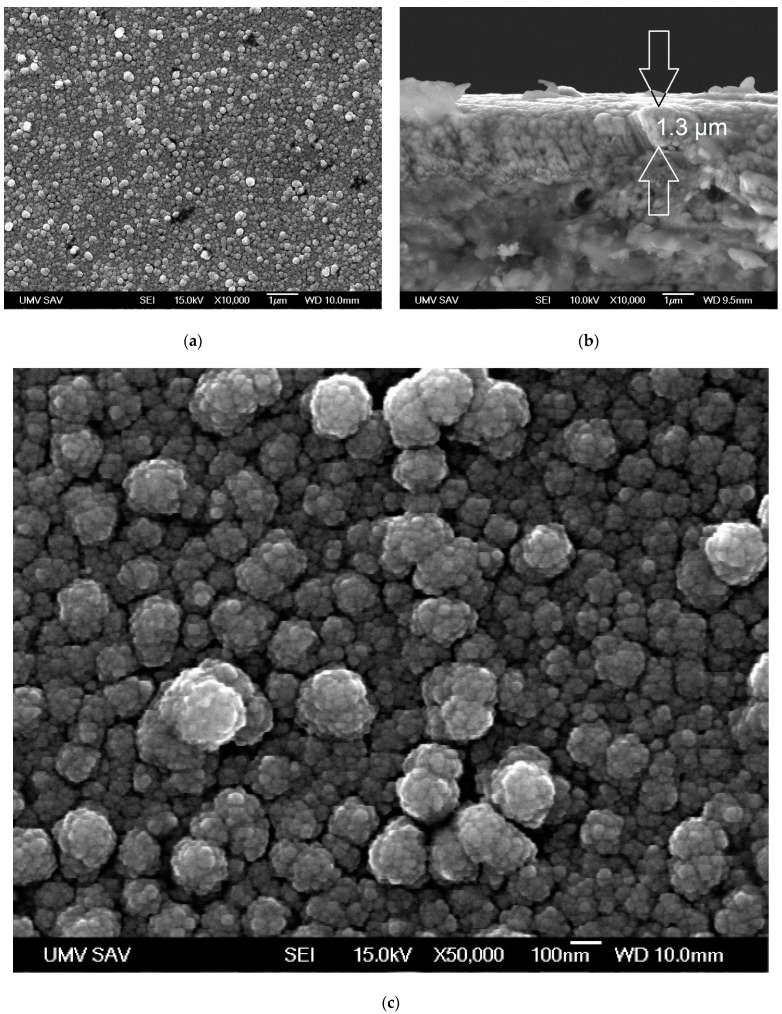
SEM images of the CrC coating: (**a**) surface morphology; (**b**) cross-sectional view; (**c**) microstructure.

**Figure 4 materials-16-05044-f004:**
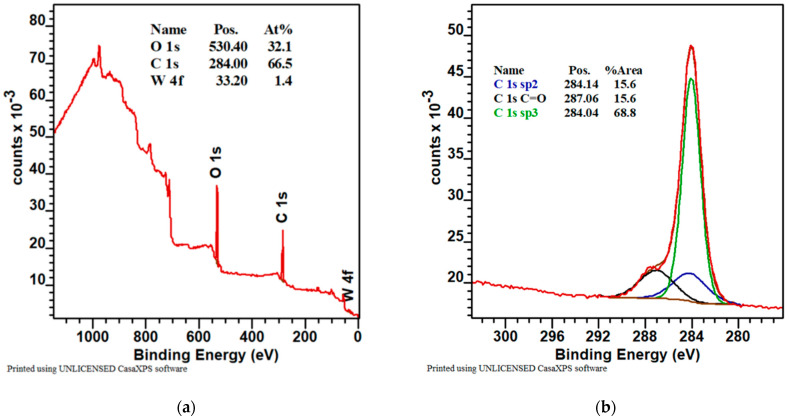
The XPS spectra of the deposited WC coating: (**a**) chemical composition (at.%); (**b**–**d**) chemical bonds of selected elements.

**Figure 5 materials-16-05044-f005:**
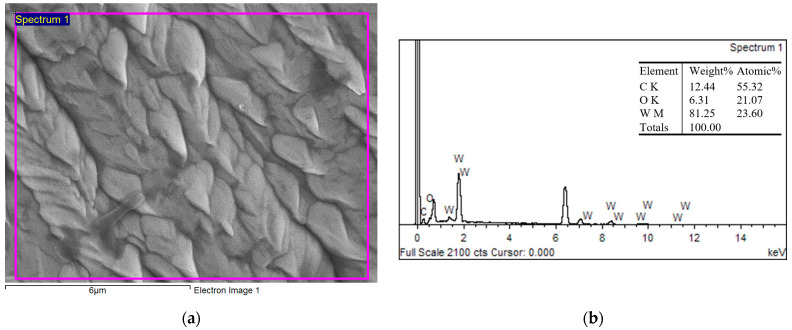
Surface morphology of the WC coating, SEM (**a**); EDS spectra of the deposited WC coating (**b**).

**Figure 6 materials-16-05044-f006:**
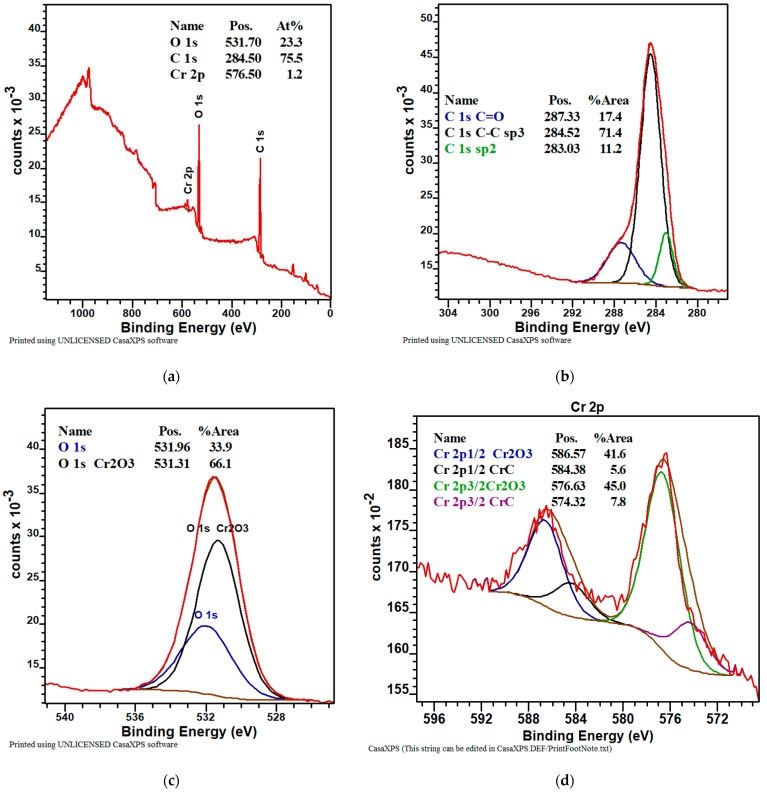
The XPS spectra of the deposited CrC coating: (**a**) chemical composition (at.%); (**b**–**d**) chemical bonds of selected elements.

**Figure 7 materials-16-05044-f007:**
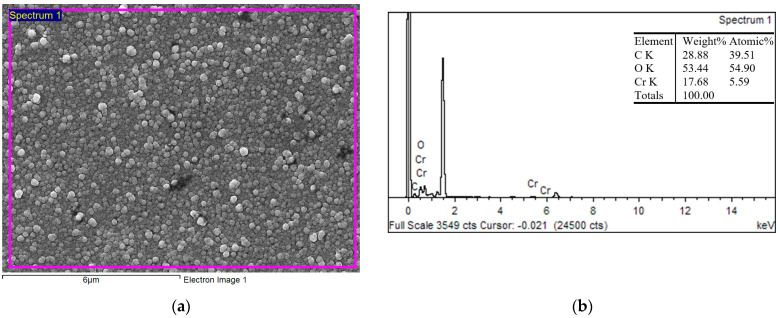
Surface morphology of the WC coating, SEM (**a**); the EDS spectra of the deposited WC coating (**b**).

**Figure 8 materials-16-05044-f008:**
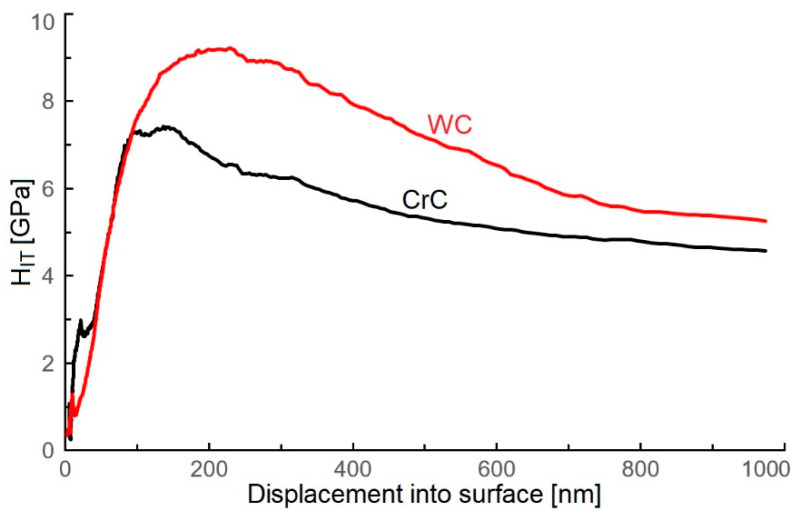
Depth profile of H_IT_ values of tested WC and CrC coatings.

**Figure 9 materials-16-05044-f009:**
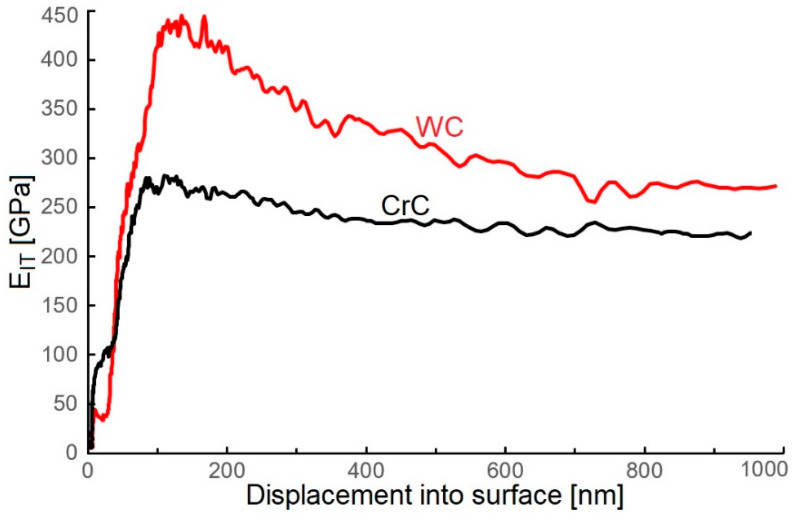
Depth profile of E_IT_ values of the tested WC and CrC coatings.

**Figure 10 materials-16-05044-f010:**
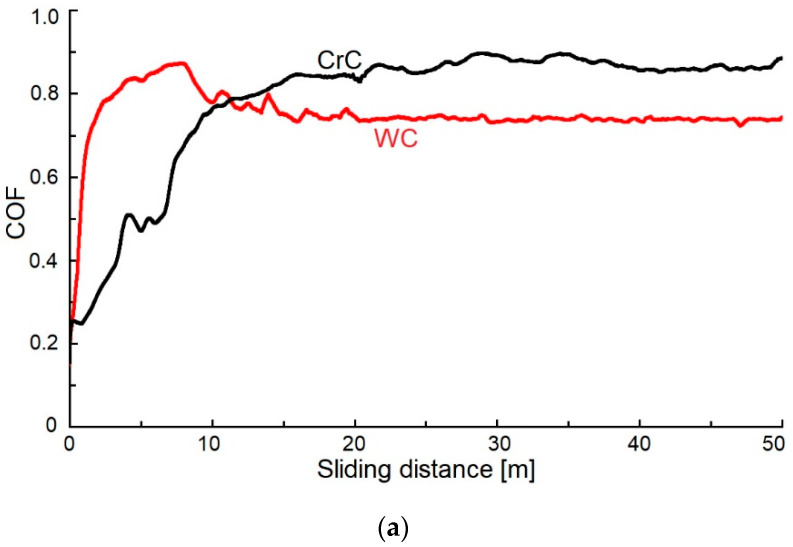
The COF values vs. the sliding distance of the tested WC and CrC coatings: (**a**) r = 4 mm; (**b**) r = 6 mm.

**Figure 11 materials-16-05044-f011:**
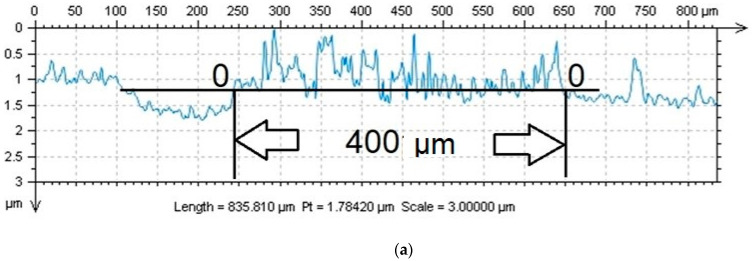
Profiles of the wear tracks after the pin-on-disc test with track area values: (**a**) WC coating, radius 6 mm; (**b**) WC coating, radius 4 mm; (**c**) CrC coating, radius 6 mm; (**d**) CrC coating, radius 4 mm.

**Table 1 materials-16-05044-t001:** The technological parameters of WC and CrC coatings deposited using the PECVD method with applications of hexacarbonyl of W or Cr as a precursor.

Working Gas	Total Pressure (Pa)	Gas Pressure (Pa)	Bias (kV)	Auxiliary Anode Voltage (V)	Deposition Time (min)	Carbonyl Weight (g)
Ar	4.0	2.0	−5.0	90	180	5.3

**Table 2 materials-16-05044-t002:** Measured values of the wear of the evaluated WC and CrC coatings after the pin-on-disc test.

Type of Coating	Radius, r (mm)	Decreased Volume (−µm^3^)	Increased Volume (+µm^3^)
WC	4	1.6 × 10^6^	3.3 × 10^6^
	6	28.6 × 10^6^	0.04 × 10^6^
CrC	4	18.84 × 10^6^	0.35 × 10^6^
	6	12.2 × 10^6^	5.9 × 10^6^

## Data Availability

Data sharing is not applicable to this article.

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
