# Peer review of "The WC and CrC Coatings Deposited from Carbonyls Using PE CVD Method—Structure and Properties"

_materials, 2023, doi:10.3390/ma16145044_

Round 1
Reviewer 1 Report
This paper compared the structure, mechanical and tribological properties of WC and CrC coatings deposited by PECVD method. However, the readers did not find any novelty in the present contribution to meet the requirement of high-impact Materials.
1. Not much in terms of uncovering new scientific knowledge/mechanisms was presented here.
2. What are the advantages of PECVD method compared with traditional PVD technology, in which the deposition temperature does not exceed 400°C.
3. The deposited coatings contains a lot of oxygen (more than 23 at.%) in XPS analysis. The author should consider the influence of oxygen when comparing the properties of different coatings.
4. The structure of WC and CrC coatings is unclear. Whether the nanocomposite structure is formed by PECVD mothod ? The XRD/TEM/Raman analysis of the coatings is missing.
5. The author clamed “Both of these coatings are comprised of a high amount of C in its hard sp3 bond form.” in line 277. Actually, the deposited coatings exhibit the low hardness. Please comment it.
Moderate editing of English language
Author Response
"Please see the attachment."

Reviewer 2 Report
The authors have studied WC and CrC coatings deposited using PECVD method from carbonyl precursors. Chemical composition, surface morphology and mechanical properties of the coatings were determined. Nevertheless, the manuscript has significant drawbacks and should not be published in the presented form. The manuscript requires major revision.
1. The manuscript was written in careless manner. First, English should be extensively revised. Some sentences are difficult for understanding, for example, lines 99-103. Second, there are many errors in the text, some of them:
line 22: point between statements is absent
line 77: write “and” instead of “a”
line 79: brackets in the formula are not needed
line 88: write “disintegration” or “decomposition” instead of “dissintegration”
line 100: write “thickness” instead of “depth”
line119: “So important” is not needed
line 126: write “working gas” instead of “gas working”
line 127: the sentence “Prior to deposition…” is not needed
lines 157, 160: letters for physical quantities should be written in Italic font
Fig. 2.: an empty square looks bad
line 205: 24.5% is Cr concentration?
line 226: “oxygen of carbon from the air”?
line 274: write “in agreement” instead of “agrement”
line 285: write “CrC” instead of “Cr”
This list is most likely not complete.
2. XPS spectra were recorded without ion beam etching, so they attributed to very thin surface oxidized, carbon-contaminated layer (3-5 nm), and absolutely unrepresentative for coating analysis in general. However, the authors incorrectly attribute XPS results to the entire coatings (lines 263, 277). In fact, the analysis of the mentioned ultrathin surface layer, as well as the description of its chemical bonds, have a very little value for the publication. It would be better to add EDS (EDX) data (with analysis depth of several hundred nm) or XPS data with ion beam etching in the revised manuscript.
3. Hardness and COF of the coatings are far from the desirable values. So, it is reasonable to find optimal PECVD process parameters that allow to obtain coatings with improved hardness and COF.
Author Response
"Please see the attachment."

Reviewer 3 Report
This study is related to deposition, characterization, and hardness and wear properties of WC and CrC coatings developed by PECVD using respective carbonyls. Authors have characterized the coatings by SEM and XPS. They have investigated hardness and wear properties of the coatings. The subject matter is appropriate for the journal. However, there are several flaws in the manuscript. It is written casually. English language is not proper. It can be published in Materials with major revisions.
Please see the attachment for comments.

English is in very bad condition. In several places, sentences are not correct, past and present tenses are mixed. Generally, experimental part should be written in past tense and other parts will be in present tense. Authors must check with a native English speaker.
Author Response
"Please see the attachment."

Round 2
Reviewer 1 Report
The changes proposed by the reviewers and editors are successfully addressed.
Minor editing of English language required
Reviewer 2 Report
Accept in present form
Reviewer 3 Report
Authors have modified the manuscript to some extent. However, there was a major concern for the analysis of XPS data which was pointed out in the first review. Unfortunately, authors have ignored the advice of the reviewer. XPS analysis is completely wrong, not as per XPS principle. C 1s component peaks are wrongly interpreted, spin-orbit peaks of different W and Cr component species in respective W 4f and Cr 2p core level spectra are not analyzed properly. It will provide wrong information to the readers about XPS binding peak positions of C 1s, O 1s, and spin-orbit peak positions of W 4f and Cr 2p peaks. As these critical issues have not been modified in the revised manuscript, the manuscript is not suitable for the publication in a journal, Materials.
Round 3
Reviewer 3 Report
It is improved now. It can be published in Materials.